# Instructional Design Made Easy! Instructional Design Models, Categories, Frameworks, Educational Context, and Recommendations for Future Work

**Hassan Abuhassna** [1,*]  **and Samer Alnawajha** [2]

1   Faculty of Social Sciences & Humanities, School of Education, Universiti Teknologi Malaysia (UTM), Skudai 81310, Malaysia
2   Faculty of Medical Sciences, Al-Aqsa University, Gaza 00972, Palestine
*   Correspondence: mahassan@utm.my; Tel.: +60-0183208713

**Abstract:** Educators and course designers may face great hurdles when designing courses if they include an online setting. Instructional design (ID) has played a vital role as a change agent in facilitating the pedagogical and technological transformation of educators and students. However, some instructors still find ID challenging and there are information gaps regarding instructional design models, categories, educational context, and recommendations for future work. This systematic literature review (SLR) analyzed 31 publications using PRISMA to address this gap. The results of this review suggest combining ID models with broader theoretical frameworks. Investigations and research on ID should include a bigger number of ID types. It is highly recommended that extra frameworks be added to the ID procedure. To explore and grasp all parties engaged in ID, including the role of the instructor, the ID designer, and the student, it is important for additional educational contexts to be amalgamated. For novices in the field, such as graduate students, it is crucial to pay close attention to the several phases and techniques of ID. This review sheds light on the trends, future agenda, and research requirements associated with ID in educational settings. It might serve as a basis for future research on ID in educational contexts.

**Keywords:** instructional design; course design; instructional technology; designing instruction

## 1. Introduction

As new technologies emerge, they bring with them new ways of imparting knowledge. Researchers are continually on the lookout for novel methods to enhance student education, particularly in online classrooms [1]. Due to the fact that certain disciplines require hands-on activities and specialized training to acquire the necessary abilities, incorporating an online environment presents educators and course designers with even more challenges [2]. This is where the importance of instructional design (ID) has grown in the educational context. In addition, instructional designers (IDs) have played a crucial role as agents of change in aiding the pedagogical and technical transformation of teachers and learners [3]. The research on quality ID for online learning has emphasized a scaffolded technique with orientation, mentoring, and continuous support [3], in addition to being receptive to instructors [4]. Furthermore, ref. [5] determined that individualized teacher training and support is one of the most important variables in assisting instructors with the transition to online education through a qualitative comparison of two methods for ID teacher training. However, the level of assistance that teachers require necessitates an extensive time commitment to assist with course development, evaluate and approve the course prior to its official release, and offer teachers continuous cooperation and support. The significant time commitment required for online education preparation is not a new discovery and has been frequently observed in the sector [6].

The term "ID" is used to describe a wide variety of positions that are derived through a combination and permutation of several distinct descriptors. Instructional designer, academic, developer, learner, engineer, educator, educational technology expert, designer, and designer of instructional technology are all terms used in this discipline [7]. This curiosity has revived old concerns about the topic of ID. It is common knowledge that defining the role of IDs is difficult [8,9]. There is uncertainty over the distinctions between the roles of IDs and instructors, as well as many other design professions, particularly graphic and multimedia design. Questions concerning the role of IDs and what they should accomplish inevitably leads to arguments concerning how they should be educated most effectively. The authors of [10,11] have highlighted how IDs should learn; they conclude that learning should include a variety of methods, including studio- and practice-based activities as opposed to more typical classroom-based methods. The authors of [12] studied skilled IDs and found unanimity on 61 ID principles; 32 of the 61 design principles fit the ADDIE framework, including understanding your students and target audience, using the best technology, and piloting, if feasible. The remaining 29 principles include communication, client management, and project management. The authors of [12] suggest that this could have a number of effects on the field of ID. IDs could undertake either case-based learning [13,14] or real-world problem solving to go beyond the model and cover the full range of principles mentioned above [15].

Consequently, the purpose of this project is to conduct an SLR that provides novel information for future researchers concerning ID models, categories, the educational context, recommendations for future work, and the research needs of ID in education. This led to the following research questions:

1. What instructional design models did previous studies employ?
2. What instructional design categories were examined in previous studies?
3. What instructional design theories did previous studies employ?
4. What instructional design frameworks did previous studies employ?
5. What educational contexts were utilized in previous research?
6. What kinds of samples were used in previous studies?
7. Where geographically were previous studies conducted?
8. What are the recommendations for potential IDs in the future?

## 2. Materials and Methods

This comprehensive SLR aims to shed some light on ID models, categories, educational context, suggestions for future study, and research requirements of ID in education for the benefit of future scholars. This SLR followed the guidelines established by the preferred reporting items for systematic reviews and meta-analyses (PRISMA) [16].

### 2.1. Exclusion and Inclusion Criteria

This study established a set of inclusion and exclusion criteria to guarantee that the selected papers fell within its scope, based on research questions derived from previously recognized research gaps. Establishing the criteria for inclusion and exclusion is essential. The inclusion and exclusion criteria for this research were developed via extensive reviews of the relevant previous literature. The inclusion and exclusion criteria for this SLR are detailed in Table 1.

**Table 1.** The inclusion and exclusion criteria.

| Inclusion Criteria | Exclusion Criteria |
|---|---|
| Instructional design, instructional technology, and course design. | Research in different environments than instructional design. |
| Articles only | Conferences, blogs, theses, and book chapters. |
| Written in English. | Any other languages. |
| The period from 2012 to 2022. | Publications from 2023 have been omitted as the year has not yet concluded. |
| Subject area (social science and arts). | Any other subject area. |

*2.2. Data Sources and Search Strategies*

In January 2023 the search for articles was undertaken. This study analyzed all publications discovered in relevant databases from 2012 to 2022; the year 2023 was removed as it is not yet complete. Based on the timing of this review, the term "PUBYEAR > 2011 AND PUBYEAR 2023" was used to provide access to the relevant publications. Scopus and Web of Science were chosen as data sources as they are 2 of the most prominent and widely used indexing organizations in the world. This SLR offers significant coverage of the academic literature on the issue at hand due to the use of a highly specific and restricted collection of keywords and search terms. The terms "course design" and "instructional technology" were both used as keywords. It resulted in the quest for knowledge. TITLE-ABS-KEY (course, design, instruction, and technology) WoS used the search terms "course design" and "instructional technology" interchangeably. In addition, for a more precise search, the option LIMIT-TO (EXACTKEYWORD, "Instructional Design") was selected from the additional capabilities offered by the Scopus database's new look search. Following a Scopus search, 131 articles were included in this study's first draught. The first WoS data search for this research found 116 papers matching the search parameters. This study evaluated a total of 247 instructional design publications published in Scopus and WoS.

The researchers then obtained the article data generated by Scopus and WoS for use in this investigation. Comparing the 2 databases revealed that 108 Web of Science articles are duplicated in Scopus. There were formerly 247 articles; however, after deleting duplicates, 131 remained. The researchers then downloaded the articles from each of the 131 publications included in our investigation. Despite several attempts, only 39 of 131 articles were accessible; other items were unavailable for various reasons, including limited access and closed access journals. For the benefit of future scholars, an analysis based on human review (manual evaluation) and specified inclusion and exclusion criteria reduced the number of papers to 39, all of which included research incorporating ID models, categories, educational context, suggestions for future study, and research requirements of ID in education. In terms of both amount and diversity, the papers presented were sufficient according to the researchers. Eight publications were discarded after being screened using inclusion and exclusion criteria for reasons such as using the phrase "instructional design" but not ID models. The incorporation of sources not often seen in academic settings was another factor. Another reason not to include a study is because it did not adequately address any component or framework of ID. In the end, 31 publications were selected for inclusion in this study. This SLR used PRISMA to analyze 31 publications, as shown in Figure 1.

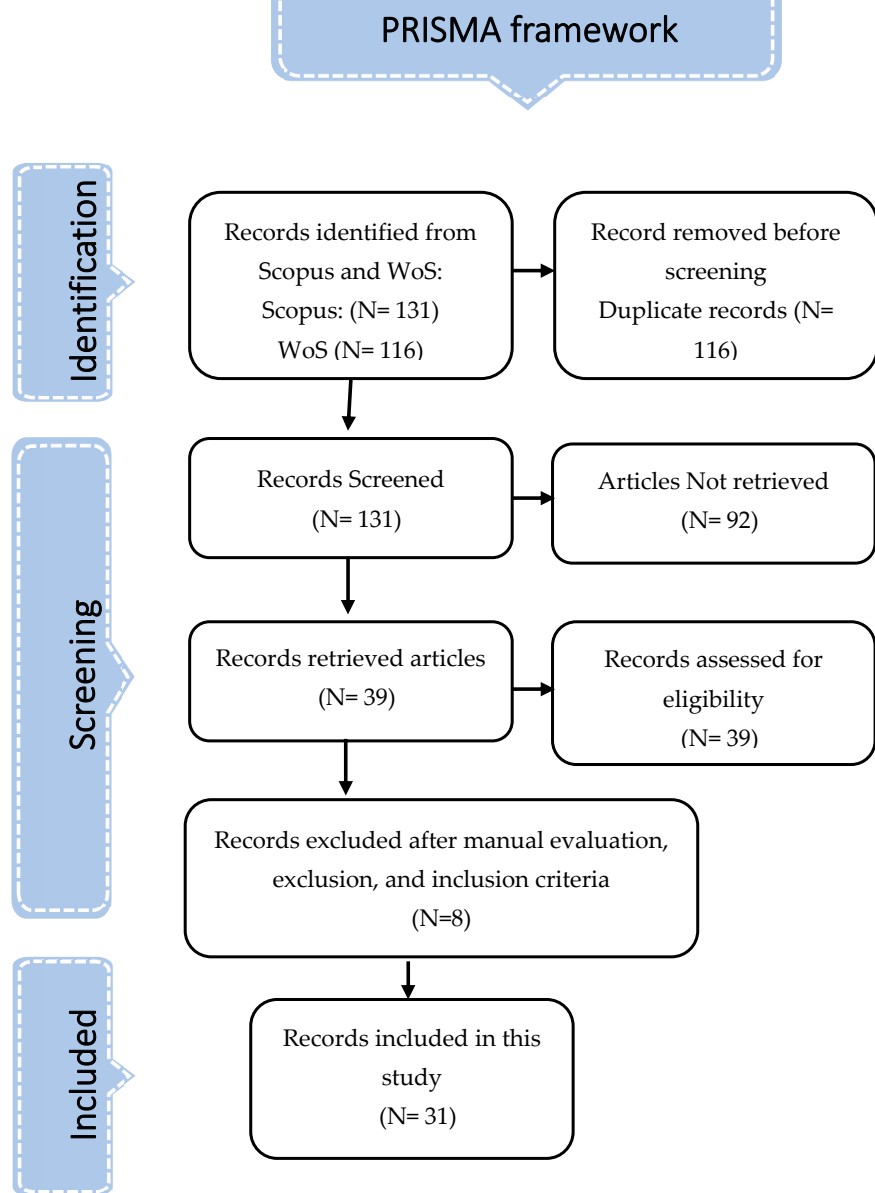

**Figure 1.** The PRISMA framework.

## 3. Results

To achieve the intended research goals, the 31 papers that were located, examined, and included in compliance with PRISMA [16] were critically and analytically analyzed to establish the direction and trends of ID in an educational environment. Appendix A of this systematic review lists the articles that were researched and included.

### 3.1. Instructional Design Models, Categories, Theories, and Frameworks

Based on the analyzed publications, there are several ID models and categories that might serve as a guide for future ID researchers and teachers in understanding and presenting the knowledge gap in the ID context [17]. These studies used ADDIE and rapid prototyping (RP). The authors of [18] utilized the technological pedagogical content knowledge-based instructional design model (TPACK-Based IDM). The authors of [19] used LITTLE in their study, which refers to learner-centered, inquiry-based, technology-enriched, trophy-driven, literature-guided, and evidence-based [20]. The authors of [21] used the sensemaking theory to study blockchain-based online education. The authors of [22] implemented a design-

based research method. In [23], the authors utilized the web pedagogical content knowledge (WPACK) model in addition to the PINTARR model, which refers to preparation, isolation, transformation, action, reflection, and revision. The authors of [24] utilized "engaging with the course content" and "communicating with the learning community" [25]. The authors of [26] used Kirkpatrick's theoretical model to study the universal design for learning (UDL) environment. The authors of [27] utilized the electronic performance support system (EPSS). In [28], the authors implemented a group of theories, including the connectivity theory, the self-directed learning theory, and the theory of the online learning community. In [29], the authors used both the teacher-centered approach (SCA) and the student-centered approach (SCA). The authors of [30] utilized social media learning activities (SMLAs) [31]. In their study, they used online-learning-related pedagogical content knowledge (PCK). The authors of [32] used the research and development (R&D) method, which consists of three main stages: system requirements analysis, system development, and formative evaluation. In [33], the authors applied cognitive load theory (CLT).

Moreover, in [34], the authors used the ADDIE model of adult learning based on adult learning theory (i.e., andragogy), teaching theory, and learning theory. In [35], the cooperative mentoring model was used. The authors of [36] implemented the interest-driven, challenge-based instructional design theory of the interest-driven creator theory. In [37], the authors utilized self-directed learning (SDL). In [38], the universal design for learning (UDL) and ADDIE were used. The authors of [39] used the 4Es learning cycle model: engagement, exploration, explanation, and extension. In [40], the authors utilized discussion forums, video lectures, supplemental readings, and practice quizzes. In [41], the mental model of instructional design (ADTRE) (analyzing, designing, teaching, revising, evaluating, and improving) was used as an instructional model. The authors of [42] used the Kemp model of instructional design. In [43], the authors utilized a few categories, including setting the stage, consistency when team teaching, timeliness in posting materials, time on task, accountability for online activities, use of structured active learning, instructor use of feedback on student preparation, incorporation of student feedback into the course, short reviews of online material during class, and ensuring technologies are user friendly.

Accordingly, the authors of [44] implemented the five aspects of instructional design: objectives, curricular content, learning activities, educational resources, and the existing evaluation strategy. The authors of [45] utilized the instructional design framework, which includes (i) examining situational factors that influence the instructional design of a course, (ii) formulating the student learning goals through course learning objectives (CLOs), and (iii) ensuring alignment of CLOs with instructional design elements. In [46], the authors used the course overview and introduction, learning objectives, assessment, instructional materials, learner interaction, course technology, learner support, and accessibility. In [47], collaborative learning was implemented by adding a module on team processes, using Google applications for communication and evaluating collaborative learning processes in addition to the products. Table 2 presents the most common ID models, strategies, and theories in relation to designing educational contexts.

Table 2 demonstrates the most often used ID models: TPACK-based IDM, WPACK, Web PINTARR, Kirkpatrick's theoretical model, ADDIE, the cooperative mentoring model, the 4Es learning cycle model, ADTRE, and Kemp's model of instructional design. Table 3 illustrates the commonly used theories in the context of instructional design.

The theories that are most often used in ID are outlined in Table 3, which are the sensemaking theory, the connectivity theory, the self-directed learning theory, the theory of the online learning community, the cognitive load theory, the adult learning theory, and the interest-driven creator theory. Table 4 presents the commonly used ID frameworks in the context of instructional design.

**Table 2.** Commonly used ID models in the context of instructional design.

| | Models | Explanation |
|---|---|---|
| [18] | TPACK-based IDM | The three-stage TPACK-based IDM teacher candidates initially learn the TPACK model. This stage prepares pre-service teachers for the design phase by helping them comprehend the TPACK model. The second step is TPACK model testing. Pre-service teachers "role play" to learn about student-centered technology integration and the TPACK paradigm. Pre-service instructors use TPACK in the final step. Pre-service teachers develop and use instructional resources to learn more about TPACK. |
| [23] | WPACK | TPACK was a prominent framework for integrating technology into ID. The TPACK framework may guide curriculum design and help theoretical and epistemological learning settings, such as web learning, to further enhance the framework to incorporate the web into a PCK known as TPACK-Web or WPACK. |
| [23] | Web PINTARR | Web PINTARR design model (preparation, isolation, negotiation, transformation, reflection, and revision). The Web-PINTARR model offers TPACK practice to novice pre-service teachers in web knowledge and competence to try as an expert adapter of web-based learning, particularly Web PINTARR, which may impact and improve pre-service teacher competencies. |
| [26] | Kirkpatrick's theoretical model | The Kirkpatrick model is an internationally known approach for assessing the effectiveness of training and education programs. It evaluates formal and informal training techniques using four levels of criteria: response, learning, behavior, and outcomes. |
| [34] | ADDIE | Creating successful learning experiences may be accomplished with the help of instructional designers and training developers that employ a learning paradigm known as ADDIE. The word "ADDIE" is an acronym that stands for a process that consists of the following five steps: analysis, design, development, implementation, and evaluation. |
| [35] | The cooperative mentoring model | This is built on mentoring principles in which the ID (the mentor) mentors the incoming university faculty (the mentee). The notion of mentoring describes the process through which individuals acquire new skills, values, and cultures directly from others they like and respect. People tend to imitate the behavior of others, particularly if it is rewarded. When individuals seek out partnerships to participate in competence-seeking activity, the notion of mentoring is likewise founded on incentive. |
| [39] | The 4Es learning cycle model | This 4E learning cycle relates to exploration, which includes laboratory activities. In this stage, the instructor guides the pupils in making connections between the findings of the activity and/or subject and previously learned material. Expansion: at this level, students are expected to use their acquired scientific knowledge. Evaluation is a crucial component in which students reflect on the material they have learned. |
| [41] | ADTRE | The ADTRE model is a visual mental model that pre-service science instructors may use to make iterative decisions in complicated and varied teaching contexts. ADTRE's five stages are: analyzing, designing, teaching, revising, and improving. Textbook, curriculum, and learner analysis define instructional tasks in the analysis phase. In the designing process, choices are made based on content, goals, tactics, media and resources, and events. Enactment is part of teaching practice. Design improves by modifying and reviewing. |
| [42] | Kemp's model of instructional design | This model's non-linear structure and interconnected components make it a novel instructional design technique. As they may start the design process with any of the nine components or phases, instructional designers have a lot of freedom. Designers do not have to examine components in an "orderly method to actualize the instructional learning systems design." |

**Table 3.** Commonly used ID theories in the context of instructional design.

| Paper | Theories | Explanation |
|---|---|---|
| [21] | The sensemaking theory | The notion of sensemaking clarifies how members of an organization perceive and make sense of shared experiences during periods of transition. Sensemaking is a collaborative process that begins at the individual level but is later altered by shared experiences and reflection. |
| [28] | The connectivity theory | The connectivity theory is implemented in a linked environment. By taking pleasure in their learning efforts, students regulate their own learning as autonomous and responsible learners. Students use a variety of strategies to manage or adapt their learning experiences to attain predetermined learning objectives. |
| [28] | The self-directed learning theory | Student learning is seen as an independent process that leads to individual accomplishments. The idea that student teachers begin and alter their learning using the flipped classroom pedagogical technique is based on self-directed learning. This technique encourages student instructors to engage in autonomous behavior. |
| [28] | The theory of the online learning community | The lecturer acts as an e-mentor and e-tutor to help students in the course when they face problems in the real world. The online learning community is used as online support for situated collaborative and cooperative learning. |
| [33] | The cognitive load theory | Cognitive load theory (CLT) is used to describe the incorporation of interactive instructional technology in higher education. The CLT emphasizes the usability of technology as an essential consideration when using multimedia as a learning aid. |
| [34] | Adult learning theory | Adult learning theories have grown to include a variety of possibilities. Self-direction, transformation, experience, mentoring, mental orientation, motivation, and learning preparedness are the seven principles of adult learning. |
| [36] | The interest-driven creator theory | The interest-driven creator (IDC) theory is a collective effort by Asian scholars to establish a comprehensive learning design theory for the future of education in Asia. The theory postulates that students may be involved in the production of knowledge (creating ideas and artefacts) when motivated by interest. |

**Table 4.** Commonly used ID frameworks in the context of instructional design.

| Paper | Framework | Explanation |
|---|---|---|
| [25–38] | The universal design for learning (UDL) environment | Universal design for learning (UDL) is a way of thinking about teaching and learning that enables all students to have an equal chance of success. This strategy provides flexibility in how students acquire content, interact with it, and demonstrate their knowledge. Universally designed learning environments (UDLE) are learning environments that accommodate a variety of students in each course. |
| [27] | The electronic performance support system (EPSS) | EPSS is a computer program that enhances user performance. It boosts productivity, quality, accuracy, turnaround time, and service while lowering training costs. Managers and workers should collaborate to establish objectives, create goals, assess performance, exchange performance reports and appraisals, and offer feedback. |
| [30] | Social media learning activities (SMLAs) | This refers to the many kinds of social media learning activities (SMLAs), the way in which they are designed, the different kinds of cognitive processes that they support, and the different kinds of knowledge that students participate in while they are completing SMLAs. |
| [45] | The instructional design framework | This means (i) looking at the situational factors that affect how a course is taught, (ii) writing down the student learning goals as course learning objectives (CLOs), and (iii) making sure that the CLOs are in line with the instructional design elements. |

In the context of ID, the frameworks that are often employed are shown in Table 3. The most common frameworks were the UDL environment, the electronic performance support system (EPSS), social media learning activities (SMLAs), and the instructional design framework. Table 5 presents the commonly used ID strategies in the context of instructional design.

**Table 5.** Commonly used ID strategies in the context of instructional design.

| Paper | Strategies |
|---|---|
| [19] | Learner-centered, inquiry-based, technology-enriched, trophy-driven, literature-guided, evidence-based (LITTLE). |
| [20] | Blockchain-based online education. |
| [25] | Engaging with the course content and communicating with the learning community. |
| [40] | Discussion forums, video lectures, supplemental readings, practice quizzes. |
| [43] | Setting the stage, consistency when team teaching, timeliness in posting materials, time on task, accountability for online activities, use of structured active learning, instructor use of feedback on student preparation, incorporation of student feedback into the course, short reviews of online material during class, ensuring technologies are user friendly. |
| [44] | Objectives, curriculum content, learning activities, educational resources, evaluation strategies. |
| [47] | Team processes, Google applications for communication. |

The most often used ID strategies may be found in the context of instructional design, as shown in Table 5. These strategies are learner-centered, inquiry-based, technology-enriched, trophy-driven, literature-guided, evidence-based; blockchain-based online education; engaging with the course content and communicating with the learning community; discussion forums, video lectures, supplemental readings, and practice quizzes; setting the stage, consistency when team teaching timeliness in posting materials, time on task, accountability for online activities, use of structured active learning, instructor use of feedback on student preparation, incorporation of student feedback into the course, short reviews of online material during class, and ensuring technologies user friendly; objectives, curriculum content, learning activities, educational resources, and evaluation strategies; and team processes and Google applications for communication.

*3.2. Variety of Educational Context*

From our analysis of the chosen articles, we can see that the ID models and categories discussed have been integrated into a wide range of educational settings. They could serve as a reference for future ID researchers and instructors in recognizing and expressing the knowledge gap in the ID setting. The vast majority of the research we looked at used online learning as its instructional setting, which was the case for five publications (16%) [24,31,34–46]. MOOCs were used as an instructional setting in the following educational contexts, accounting for four publications (13%) [19,37,38,40]. Furthermore, four publications (13%) [17,29,39,43] in competition with MOOCs used blended learning for teaching and learning purposes. Three studies (10%) conducted their instruction in a face-to-face setting [22,36,41]. Furthermore, the remaining research employed a wide range of pedagogical settings, with 15 publications (48%) utilizing the following: Ref. [18] online education platforms for e-learning, ref. [21] virtual learning environments, ref. [23] courses integrated with the web, ref. [25] LMS through an online lab course, ref. [26] simulation-based training, and [27] Moodle and flipped instructional design (FID). The authors of [28] used LMS platform and flipped instructional design, while the authors of [30] used learning management systems (LMS). The authors of [32] used an internet-based interactive learning system. In [33], interactive instructional technology was used. The authors of [42] used interactive e-books; the authors of [44] used face-to-face and blended courses, and in [45],

flipped learning environments were used. Finally, the authors of [47] used virtual teams. Figure 2 shows the ID models and educational settings.

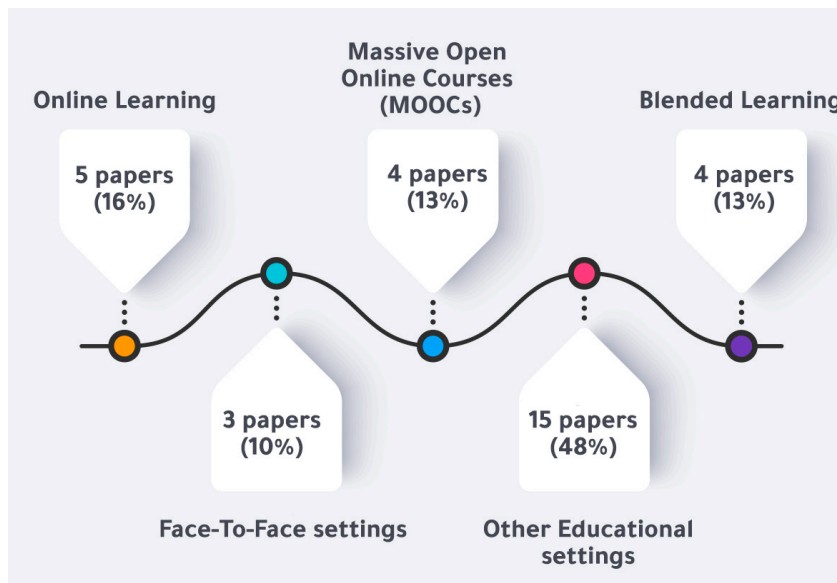

**Figure 2.** ID models and educational settings.

*3.3. Type of Samples*

Understanding the kind of sample is important for justifying the selection of samples for future research and understanding the present knowledge gap in the ID setting. ID research is virtually exclusively undertaken on humans. Based on our examination of the samples used in the selected publications, we can say with confidence that the vast majority of samples for ID studies consist mostly of "learners" [24–27,29,30,32,33,37–39,43–45,47]. Fifteen ($n$ = 15) total samples were collected, with learners accounting for 48% of the total. Only 16% ($n$ = 5) of the samples studied were "preservice teachers" [18,23,28,36,41]. In addition, 13% ($n$ = 4) of the studies surveyed "students and teachers." Moreover, 6.5% ($n$ = 2) of "instructional designers" were surveyed. Moreover, 16% ($n$ = 5) differed across various samples, for example: Ref. [21] faculty-training courses, ref. [35] university faculty member, ref. [22] educators and learning designers and technology specialists, ref. [31] Pedagogy experts, and [40] MOOC instructors. Figure 3 depicts the sample distribution of the articles analyzed.

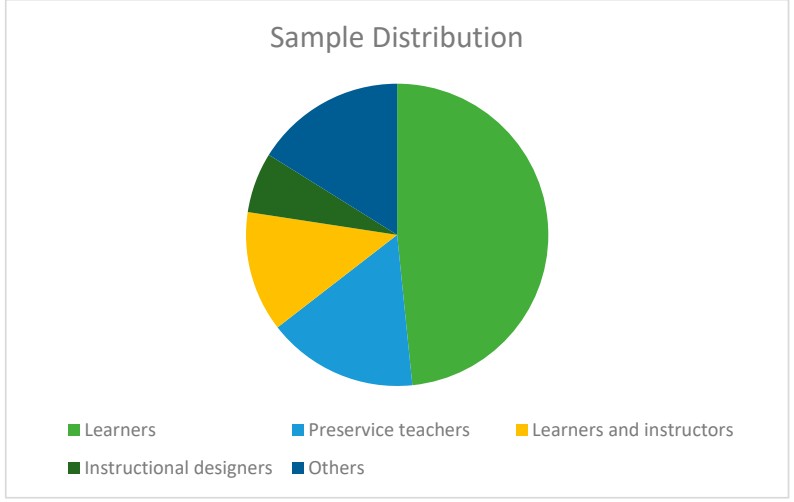

**Figure 3.** ID sample distributions.

### 3.4. Geographical Locations

The study of ID in an educational context is geographically diverse, so there is no particular emphasis on places. There are, nonetheless, clear indications of high scientific activity in the USA, with 13 articles representing 42% of the total publications. Studies [19,21,24,25,33–35,37,39,40,43,45,47] were all carried out and published in the US. There have been just two investigations undertaken in Turkey [18–27], two more in Australia [22–31], and two in Indonesia [23–32]. In the interim, 12 papers were published globally, accounting for 39% of the articles examined: Ref. [17] Bangladesh (*n* = 1), ref. [20] India (*n* = 1), ref. [26] Uganda (*n* = 1), ref. [28] South Africa (*n* = 1), ref. [29] France (*n* = 1), ref. [30] Lebanon (*n* = 1), ref. [36] Malaysia (*n* = 1), ref. [38] Colombia (*n* = 1), ref. [41] China (*n* = 1), ref. [42] Mexico (*n* = 1), ref. [44] Costa Rica (*n* = 1), and [46] Canada and Spain (*n* = 1). Figure 4 depicts the geographical distribution of TDT and distance learning publications.

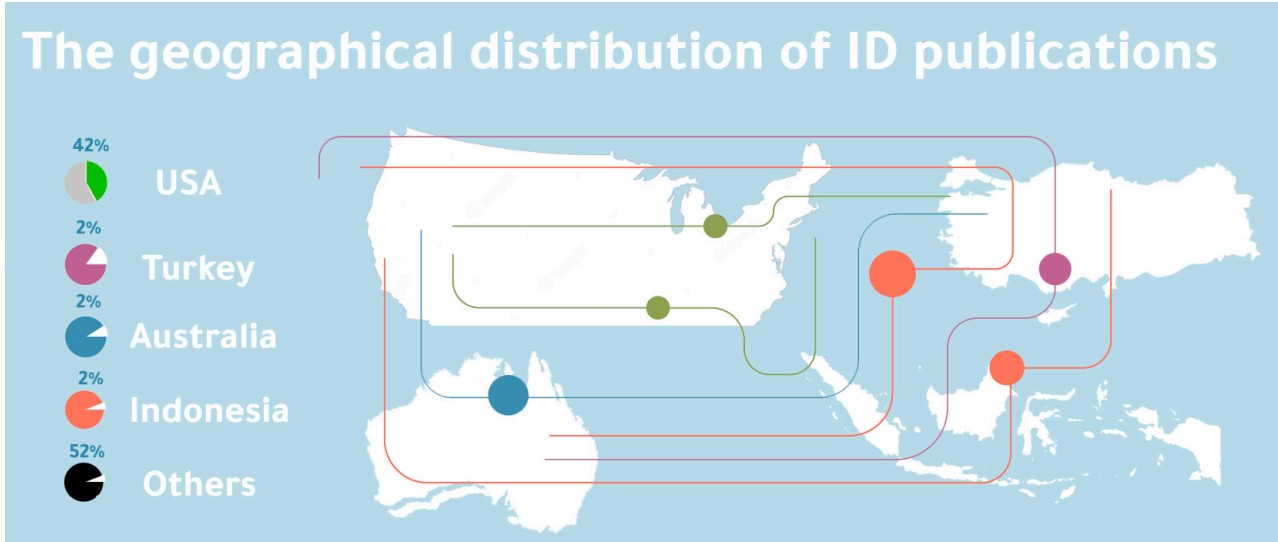

**Figure 4.** Geographical distribution.

### 3.5. Recommendations for Future ID

From what we can gather from the reviewed literature, the most widely accepted piece of advice is that instructional designers might encourage professors to guide submodules based on expertise and experience or lead small peer groups inside broader training sessions [21]. There are fundamental design concepts (many entry and exit points, flexibility in course delivery, proximity to practice, prioritizing communication and experience over knowledge) in order to build a comprehensive framework for learning design instruction [22]. Social media affordances may engage students' higher cognitive processes and knowledge in SMLAs [30]. Online instruction is an important aspect of such professional preparation. Now more than ever, universities should invest in faculty professional development to update them on successful teaching approaches with or without online tools [31]. Students, instructors, and course material are separated by technology. Faculty must be acquainted with technology to develop trust with students, while students must handle technical issues [33]. Instructional designers possess relevant knowledge and are motivated to learn more to advance their positions and talents. More significantly, instructional designers know what does not work in their field and believe that administrative and project work hinders skill development and career progress [34]. Further study is needed to assist IDs as mentors and university professors using institutional structures and methods. Why academics fail to finish mentoring despite institutional support needs further investigation (a grant, in our case). Finally, mentorship may help experienced online educators, but it must be tailored to them [35].

## 4. Discussion

The construction of educational materials is referred to as ID. This profession, however, goes beyond just developing instructional materials; it carefully evaluates how students learn and what resources and approaches would most successfully assist people in achieving their academic objectives. IDs use learning theory and a systematic approach to create material, learning activities, training, and other solutions to improve the teaching and learning process. Furthermore, this review offers some insight into the state of current ID research and what future research should consider. As a result, the purpose of this SLR is to understand how ID has enhanced our understanding of learning contexts. Based on this study, specialists in the field will have a clearer understanding of where ID is headed and will be aware of research gaps that may be exploited to initiate new projects.

### 4.1. Instructional Design Models, Categories, Theories, and Frameworks

This SLR revealed several interesting and significant facts. As seen by the papers we examined, most of the ID research incorporates aspects of various models and frameworks.

However, when examining the learning theories used in these investigations, there seems to be an increasing trend toward avoiding integrating ID with other learning theories. For instance, refs. [21,28,33,34,36] were theoretically based. When investigating ID frameworks in educational contexts, we discovered that [25–38] made use of the UDL environment. This is because the design strategies of this kind of online course may be improved by addressing UDL, accessibility, usability, and online AT access. Our method not only improves access to school for children with functional variety, but it also enhances the overall learning experience. Another study [27] employed EPSS because it increased students' academic performance and met students' expectations. They will use similar solutions in the future since the technology worked effectively. SMLAs, on the other hand, were used because social media affordances may engage students' higher cognitive processes and knowledge in SMLAs. Finally, ref. [45] stated that a low-cost, modified flipped model was established by examining situational concerns, revising course learning objectives, and incorporating instructional design. Students' grasp of electrical and computer engineering applications improved because of team-based collaborative and active learning activities, as did their critical thinking and problem-solving skills.

Furthermore, by discussing ID models in educational contexts according to the sample examined, the integration of ID models was used in many research projects to study certain external aspects directly associated with students in learning settings. For example, refs. [18,23,26,34,35,39,41,42] each used a broad range of models in their research, which led them to understand why some training programs work and others do not. Future research should consider this method. Furthermore, other elements influenced this integration, such as the extensive knowledge of instructional designers and their eagerness to learn more to progress their positions and abilities. More importantly, instructional designers understand what does not work in their area and believe that administrative and project work impedes skill development and career advancement.

Finally, based on our review of the literature, we have found that ID research has embraced a broad variety of ID strategies, such as the one described in [19], (LITTLE), which refers to learner-centered, scientific method, inquiry, technology, trophies, literature, and evidence. Online courses are secured by the blockchain [20]. Other ID strategies include participating actively in the material covered in class and sharing information with the academic group [40] and online chat rooms, video classes, supplementary materials, and sample exams [43]. Efficient ID strategies can provide context, foster harmony among the faculty, increase rapidity in uploading content, increase work time, and help to foster a sense of responsibility for one's online actions. This is achieved through utilization of active learning structures, utilization of student preparation feedback by teachers (including student comments in course development), technology-based classrooms that include frequent, brief evaluations of internet content that is fit for purpose, subject matter, instructional

strategies, assessments, and materials [44]. A team approach and Google's messaging suite was used to measure performance [47].

In sum, based on our findings, we suggest that future studies of ID look at student autonomy, satisfaction, and instructor–student interactions by combining ID models with other theories and frameworks, such as the transactional distance theory (TDT).

### 4.2. Variety of Educational Context

We found a broad variety of educational settings used in ID research among the publications we examined. The vast majority of the research we looked at used online learning as its instructional setting, followed by MOOCs and blended learning in a face-to-face setting. Further, other research employed a wide range of pedagogical settings, including online education platforms for e-learning, virtual learning environments, courses integrated with the web, LMS through an online lab course, simulation-based training, Moodle and flipped instructional design (FID), LMS platform and flipped instructional design, learning management systems (LMS), an internet-based interactive learning system, interactive instructional technology, interactive e-books, face-to-face and blended courses, flipped learning environments, and virtual teams.

Our suggestions for future research are to utilize more integration between different educational settings, as we have noticed that only one study integrated face-to-face and blended courses.

### 4.3. Type of Samples

Most relevant research included humans as ID is concerned with social and communication processes. Most of the articles we read included "students" as their participants. The study must include a diverse group of participants from the same distance-learning context. While the thoughts and opinions of students are important in educational settings, so too are examples from teachers and those who design the curriculum. In ID research, only two evaluations had been undertaken, so no firm conclusions could be drawn concerning the future of using professors as samples. However, this knowledge gap may act as a springboard for more research and the identification of new fields of study on ID in academic settings.

### 4.4. Geographical Locations

Studies have mostly focused on the United States and other developed western economies. Therefore, we conclude that the infrastructures of these nations allowed for the extensive implementation of ID research. As a result, concerns regarding the prevalence and accessibility of ID research are reduced. Therefore, the motivating viewpoints of clients are given the greatest consideration. According to the selected literature, Africa and Asia are underexplored relative to other locations, such as the United States, and should be highlighted as areas of research importance. In Asia, several emerging countries have conducted ID research in the past, including Lebanon and Malaysia. As such, the field is predicted to continue to thrive.

### 4.5. Recommendations for Future Researchers

In our experience, evaluations of forthcoming works have been less thorough. The recommendations for further educational ID research are as follows:

The correct use of course technology, learner support, and accessibility are important for great design.

Collaborative and active learning activities have boosted students' understanding. In higher education assessment, regardless of teaching medium, all instructional design variables must be included.

There is an obvious and current need to conduct follow-up, in-depth inquiries with MOOC instructors on their real instructional design approaches.

The purpose of any technology integration is to enable students to become independent learners and active participants in their own education.

Technology played a central role in supporting students' self-monitoring.

Students, instructors, and course material are separated by technology.

Faculty must be acquainted with technology to develop trust with students, while students must handle technical issues.

Online technology makes it feasible to construct interactive-based instructional designs.

Online instruction is an important aspect of such professional preparation. Universities should invest in faculty professional development to update them on successful teaching approaches with or without online tools.

To understand why some training programs work and others do not, future research should evaluate the instructional design of a training program.

To explore amateur and expert instructors' instructional design competency over time, further research should be undertaken.

## 5. Future Directions and Research Gaps

This SLR covers ID models, theories, frameworks, sample types, locations, and the future recommendations of ID. Based on a thorough and rigorous analysis of 31 articles using PRISMA, the following future actions and research requirements have been identified. The agenda and research gaps are detailed in Table 6.

**Table 6.** Future directions and research gaps.

| Component | Future Agenda | Research Gap |
|---|---|---|
| Instructional design models | Combining ID models with wider theoretical frameworks. | There is insufficient research examining the relationship between ID and other frameworks. |
| Instructional design categories | Investigations and studies pertaining to ID should include a greater number of ID categories. | Insufficient attention is given to ID categories to participate in distance learning. |
| Instructional design theories | It is recommended that other learning theories be included in the ID process, such as the transactional distance theory. | Very few theories have been included in the research that has been evaluated. |
| Instructional design frameworks | It is strongly suggested that additional frameworks be included in the identification process. | Very few theoretical frameworks have been included in the research that has been evaluated. |
| Educational contexts | More educational settings need to be combined so that everyone involved in ID, such as the teacher, the ID designer, and the student, can be studied and understood better. | There are few studies that combine many types of educational settings into a single investigation, such as online learning and traditional classroom instruction. |
| Geographical area | The United States is becoming an increasingly significant location for ID research. | More research is required in Asia, Africa, South Africa, and Europe. |
| Recommendation and future work | Those who are new to the field, such as postgraduate students, need to pay close attention to the many steps and processes of ID. | The use of several research models, frameworks, and theories will contribute to the expansion of ID practices. |

## 6. Limitation

This study looked at several instructional design models and categories, as well as educational settings and recommendations for further work. During our inquiry, we combed through the Scopus and WoS databases to retrieve relevant data. Nevertheless, there were several obstacles to overcome over the course of this inquiry. Firstly, not all extracted articles were able to be recovered, and the final number of publications to be examined was only 31. As we were unable to broaden our research, this may have influenced the results. Secondly, making use of both Scopus and WoS resulted in almost the same number of publications, with just a small handful of articles (approximately two or three) being added to the Scopus database.

## 7. Conclusions

In the realm of education, instructional design (ID) is a fundamental concept that has garnered a lot of attention in recent years. As public awareness of online education increased, so did the need for more information on the field's most recent advancements, prospective future lines of investigation, and current information voids. This SLR assessed papers based on their models, categories, theories, frameworks, contribution to educational settings, sample size, geographical emphasis, and opportunities for future research. A future study is proposed to combine ID with other theories and models and engage a mixed sample of lecturers and students. With the support of studies conducted on several continents, such as Africa and Asia, the scientific community may obtain a more complete and nuanced knowledge of ID in education. This SLR summarized the potential integration of ID with various theories. It is recommended that ID research includes a greater variety of theories and frameworks, as well as elements such as the extent to which students seek out distant learning contexts. Future research in Asia, Africa, and Europe is anticipated to contribute to the global understanding of ID and its applicability in designing learning contexts.

**Author Contributions:** H.A.; formal analysis, H.A. and S.A.; methodology, H.A.; project administration, H.A.; validation, H.A. and S.A.; visualization, H.A.; writing—original draft, H.A.; writing—review and editing, H.A. and S.A. All authors have read and agreed to the published version of the manuscript.

**Funding:** This research received no external funding.

**Institutional Review Board Statement:** Not applicable.

**Informed Consent Statement:** Not applicable.

**Data Availability Statement:** The data presented in this study are available upon request from the corresponding author.

**Conflicts of Interest:** The authors declare no conflict of interest.

## Appendix A

**Table A1.** List of sampled articles.

| Label | Article | ID Model or Categories | Educational Context | Location | Samples | Recommendation for Future Work |
|-------|---------|------------------------|---------------------|----------|---------|--------------------------------|
| A1 | [17] | ADDIE and rapid prototyping (RP) | Blended learning | Bangladesh | Students and instructors | In blended learning for polytechnic students, instructional design is a reliable and valid pedagogical strategy. |
| A2 | [18] | Technological pedagogical content knowledge-based instructional design model (TPACK-based IDM) | Distance learning | TURKEY | Pre-service teachers | The TPACK-based IDM boosted pre-service teachers' TPACK by linking their technical, pedagogical, and content knowledge. |
| A3 | [19] | Learner-centered, inquiry-based, technology-enriched, trophy driven, literature-guided, and evidence-based (LITTLE). | MOOC | USA | Students and instructors | MOOC instructors and designers may cater to teachers' greatest needs. The post-course survey may also enhance future course design. |
| A4 | [20] | Blockchain-based online education | E-learning platforms | India | Students and instructors | To ensure security and integrity and to avoid fake evaluations and rankings of online courses, an immutable ledger, a decentralized, transparent, and safe rating method, and the latest technology to follow the course at every level are needed. |
| A5 | [21] | Sensemaking theory | Virtual learning environment | USA | Faculty-training courses | Instructional designers might encourage professors to guide submodules based on expertise and experience or lead small peer groups inside broader training sessions. |
| A6 | [22] | A design-based research method | Learning design content | Australia | Educators and learning designers, technology specialists | The fundamental design concepts (many entrance and exit points, flexibility in course delivery, near to practice, and prioritizing communication and experience over knowledge) to construct a complete and thorough framework for learning design instruction. |
| A7 | [23] | Web pedagogical content knowledge (WPACK), the preparation, isolation, transformation, action, reflection, and revision (PINTARR) model | Courses integrated with the web | Indonesia | Pre-service teachers | Pre-service teachers must understand web design since integrating technology into teaching and learning is complicated. Thus, the teacher suggested altering a new technological integration experience. |
| A8 | [24] | Engaging with the course content, communicating with the learning community | Mobile device use in online learning | USA | Learners | Online students may struggle to access other online platforms and be diverted from academic work, limiting the potential benefits of mobile device usage. |
| A9 | [25] | Universal design for learning (UDL) environments | LMS online lab course | USA | Learners | The online laboratory course may be given in 6-, 8-, or 16-week segments throughout the year. Many students struggle to obtain online lab courses due to demand. Course design maximized student access and experience using UDL, genuine learning environment design, and accessibility best practices. |

**Table A1.** *Cont.*

| Label | Article | ID Model or Categories | Educational Context | Location | Samples | Recommendation for Future Work |
|-------|---------|------------------------|---------------------|----------|---------|-------------------------------|
| A10 | [26] | Kirkpatrick's theoretical model | Simulation-based training | Uganda | Learners | A technology-enhanced simulation-based obstetrics training in a low-income nation scored well, although intervals were substantial. To understand why some training programs work and others do not, future research should evaluate the instructional design of a training program. |
| A11 | [27] | Electronic performance support system (EPSS) | Moodle flipped instructional design (FID) | Turkey | Learners | This study's mid-level EPSS improved students' scientific research techniques academic performance and matched student expectations. The technology performed well, and they will utilize similar solutions in the future. |
| A12 | [28] | The connectivity theory and self-directed learning theory and online learning community. | Flipped instructional design and LMS platform | South Africa | Preservice teachers | The flipped instructional design (FID) modeled, supported, and actively engaged student teachers in the approach for personal benefits. Student instructors found flipped instructional design fascinating and beneficial to their learning. |
| A13 | [29] | Teacher-centered approach (SCA) Student-centered approach (SCA) | Blended learning environments | France | Learners | Three of seven student-centered blended learning courses greatly increase students' self-directed learning. The data also suggest that lecturers of students who increased their self-directed level offered online peer review and forum discussion activities. |
| A14 | [30] | Social media learning activities (SMLAs) | Learning management systems (LMS). | Lebanon | Learners | Social media affordances may engage students' higher cognitive processes and knowledge in SMLAs. |
| A15 | [31] | Online learning-related pedagogical content knowledge (PCK) | Online learning environments | Australia | Pedagogy experts | Online instruction is an important aspect of professional preparation. Now more than ever, universities should invest in faculty professional development to update them on successful teaching approaches with or without online tools. |
| A16 | [32] | The research and development (R&D) method, consisting of three main stages: system requirements analysis, system development, and formative evaluation. | Computer-assisted instruction (CAI) model with a web-based interactive learning system | Indonesia | Learners | CAI-based instructional design using the tutorial, drill, and practice models may drive students to study database systems independently. Online technology makes it feasible to construct interactive CAI-based instructional designs that are linked to external learning resources that give a range of relevant materials to augment students' learning ideas. |
| A17 | [33] | Cognitive load theory (CLT) | Interactive instructional technology | USA | Learners | Students, instructors, and course material are separated by technology. Faculty must be acquainted with technology to develop trust with students, while students must handle technical issues. |
| A18 | [34] | (ADDIE) model, adult learning models based on adult learning theory (i.e., andragogy), teaching theory, and learning theory. | Online learning | USA | Instructional designers | Instructional designers have the required knowledge and are motivated to learn more to advance their positions and talents. More significantly, instructional designers know what does not work in their field and believe that administrative and project work hinders skill development and career progress. |

**Table A1.** *Cont.*

| Label | Article | ID Model or Categories | Educational Context | Location | Samples | Recommendation for Future Work |
|---|---|---|---|---|---|---|
| A19 | [35] | THE cooperative mentorship model | Asynchronous online course | USA | University faculty member | Further study is needed to assist IDs mentor university professors using institutional structures and methods. Why The reason why academics fail to finish mentoring despite institutional support needs further investigation (a grant, in our case). Finally, mentorship may help experienced online educators but it must be tailored to them. |
| A20 | [36] | Interest-driven creator theory, interest-driven, challenge-based instructional design. | Traditional classes | Malaysia | Pre-service teachers | This research implies that the InDeC learning design framework might improve educational technology courses. IDC theory and the challenge-based learning framework together yielded better results than either alone. |
| A21 | [37] | Self-directed learning (SDL) | MOOCs | USA | Learners | MOOC instructors, teaching assistants, and peers provided external feedback for students' self-monitoring. From these findings, technology played a central role in supporting students' self-monitoring. |
| A22 | [38] | The universal design for learning (UDL) and ADDIE | MOOC | Colombia | Learners | The design techniques for this sort of online course may be enhanced by addressing UDL, accessibility, usability, and online AT access. Our program not only increases access to school for children with functional diversity, but also improves the learning experience for everyone. |
| A23 | [39] | 4Es learning cycle model: engagement, exploration, explanation, and extension. | A webinar integration tool for blended environments. | USA | Learners | Adopting more student-centered approaches to learning with technology is the most effective way to engage students in webinar blended learning. Ultimately, the purpose of any technological integration is to enable students to become independent learners and active participants in their own education. |
| A24 | [40] | Discussion forums, video lectures, supplemental readings, and practice quizzes. | MOOC | USA | MOOC instructors | There is an obvious and current need to conduct follow-up, in-depth inquiries with MOOC instructors on their real instructional design approaches, especially the ways through which individualized learning is attempted and any instructional adjustments and adaptations performed over time. |
| A25 | [41] | The mental model of instructional design ADTRE (analyzing, designing, teaching, revising, and evaluating/improving) instructional model. | Face-to-face | China | Pre-service teachers | Other instructional design skill assessments may be investigated in future research. They might also cover chemistry, physics, earth science, and in-service college and university professors. To explore rookie and expert instructors' instructional design competency with time, further research may be undertaken. |
| A26 | [42] | Kemp model of instructional design | Interactive e-books | Mexico | Professors and students | The academics noted that interactive e-books helped pupils learn technology, reading, writing, cognition, and metacognition. Due to its audio and visual features, students said interactive e-books improve their marks. |

**Table A1.** *Cont.*

| Label | Article | ID Model or Categories | Educational Context | Location | Samples | Recommendation for Future Work |
|---|---|---|---|---|---|---|
| A27 | [43] | Setting the stage, consistency when team teaching, timeliness in posting materials, time on task, accountability for online activities, use of structured active learning, instructor use of feedback on student preparation, incorporation of student feedback into the course, short reviews of online material during class, and ensuring technologies are user friendly. | Blended learning | USA | Learners | Blended learning instructors should include these best practices into course design and administration. These techniques should be tested to determine whether they improve student achievement. |
| A28 | [44] | The five aspects of instructional design: objectives, curricular content, learning activities, educational resources, and the existing evaluation strategy. | Face-to-face and blended courses | Costa Rica | Learners | In higher education assessments, regardless of teaching medium, all instructional design variables must be included. After the training, indicators are needed to assess quality and address any issues. |
| A29 | [45] | Instructional design framework includes (i) examining situational factors that influence the instructional design of a course, (ii) formulating student learning goals through course learning objectives (CLO), and (iii) ensuring alignment of CLOs with instructional design elements. | Flipped learning environments | USA | Learners | A low-cost modified flipped model was created by assessing situational considerations, updating course learning goals, and integrating instructional design. Team-based collaborative and active learning activities have boosted students' understanding of electrical and computer engineering applications and strengthened their critical thinking and problem-solving abilities. |
| A30 | [46] | Course overview and introduction, learning objectives, assessment, instructional materials, learner interaction, course technology, learner support, and accessibility. | Online learning | Canada Spain | Instructional designers | Designers considered all eight categories—course overview, alignment of learning goals, assessment procedures, current instructional materials, effective learner interaction, correct use of course technology, learner support, and accessibility—important for great design. |
| A31 | [47] | Collaborative learning, adding a module on team processes, using Google applications for communication, and evaluating collaborative learning processes in addition to the products. | Virtual teams | USA | Learners | Students learned how to work together, built professional connections, got more involved, and appreciated the instructor's facilitation. |

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
