# Peer review of "Instructional Design Made Easy! Instructional Design Models, Categories, Frameworks, Educational Context, and Recommendations for Future Work"

_ejihpe, doi:10.3390/ejihpe13040054_

Round 1

Reviewer 1 Report

Dear Authors,

The paper is clearly written and properly structured. All text is comprehensive, and tables/figures are useful and readable. I believe that the paper add knowledge to the research field, is original, with recent references. The length of the paper is appropriate, and I think that the paper contains relevant information.

My decision is accepting the manuscript after minor changes:

·         Figure 2 is not named in the text.

·         Really, were only 39 of 131 things accessible? Did you check the personal webpages of the authors?

·         Line 112, when you said “various reasons….. and others”, what do you mean? What reasons?

Also, I think that you could improve your paper if you make an analysis taking into account the subject (that students are learning or teacher are teaching) or the level (primary, high school, university, etc..)

Regards.

Author Response

thank you for the comments 

we have addressed all comments as attached. 

Reviewer 2 Report

Despite the interesting content analysis, the article analyzes a small number of scientific articles. In addition, I would like to note that the article would be better if there were more author's conclusions about the characteristics and advantages of a particular model. Readers would also be interested to see recommendations for implementing models in different pedagogical settings.

Author Response

thank you for the comments 

the response is attached.  
